# Machine learning climbs the Jacob's Ladder of optoelectronic properties

Malte Grunert ✉, Max Großmann & Erich Runge

The use of machine learning (ML) as a powerful tool for the prediction of optoelectronic properties is still hampered by the inadequate level of the calculated training datasets, which are almost exclusively obtained within the independent-particle approximation (IPA). Drawing on Perdew's Jacob's ladder analogy in density functional theory, we demonstrate how ML can ascend from the IPA to the random phase approximation (RPA), figuratively climbing the second rung. We show that as few as 300 RPA calculations suffice to fine-tune a graph attention network initially trained on 10,000 IPA calculations. Its prediction accuracy approaches that of a network directly trained on our large database of around 6000 RPA spectra. Our results highlight how transfer learning even with a small amount of high-fidelity data significantly improves predicted optical properties. Moreover, by retraining on RPA data from materials with smaller unit cells, the model generalizes effectively to larger unit cells, demonstrating broad scalability.

Understanding and predicting the optical and optoelectronic properties of matter via excited-state calculations is crucial for a range of highly relevant technological applications, from photovoltaics to optical data processing and possibly quantum computing. One can think of the invoked approximation schemes as a ladder, analogously to the Jacob's Ladder in electronic structure theory based on density functional theory (DFT) as popularized by John Perdew and Karla Schmidt[1]: This conceptual tool classifies increasingly realistic, yet computationally more expensive ground-state methods.

In general, ab initio theoretical predictions of optical and optoelectronic properties comprise two steps: (i) The electronic band structure and dipole matrix elements are determined by calculating the energies and single-particle wavefunctions of the system. (ii) These energies and wavefunctions are used to calculate the optical properties, i.e., the dielectric function, which describes optical responses such as absorption and reflectivity, in what can essentially be considered as a more or less costly post-processing step. In both steps, different approximations can be made and mixed in practical workflows. The Jacob's Ladder of electronic structure theory, i.e., aspect (i), ascends from (semi-)local DFT functionals, such as the local density approximation (LDA) or the generalized gradient approximation (GGA), to non-local meta-GGA or hybrid functionals[2]. Many-body-

perturbation theoretical (MBPT) approaches such as the *GW* approximations and beyond or dynamical mean-field theory for systems with strong on-site correlations are alternatives to classical DFT methods. Regarding the evaluation of optoelectronic properties, i.e., of the dielectric function in step (ii), the lowest rung (i.e., the simplest approach) is second-order perturbation theory in the independent-particle approximation (IPA) using Kohn-Sham eigenstates[3,4]. The next rung is the random phase approximation (RPA)[5], which includes local-field effects, screening, and plasmonic excitations. We note that we follow the naming convention of YAMBO[6,7] regarding IPA and RPA etc., see below. The third rung is defined by the solution of the Bethe-Salpeter equation (BSE)[8], which most importantly includes excitonic effects. A good comparison of these methods and agreement to experimental spectra is available for example in Ref. 9, although, as is common in the literature, only a few prototypical systems are analyzed there. Unfortunately, including more and more many-body effects in either (i) or (ii) makes the calculations dramatically more expensive, precluding practical calculations including all of them for all but the smallest systems. This problem remains even when time-dependent DFT (TDDFT) is invoked[10–12], which de facto reduces the 4-point functions of MBPT to much cheaper 2-point functions[3]. To the best of our knowledge, the best performing TDDFT kernels explicitly reconstruct

Institute of Physics and Institute of Micro- and Nanotechnologies, Technische Universität Ilmenau, Ilmenau, Germany.
✉e-mail: malte.grunert@tu-ilmenau.de

the BSE at comparable computational cost, with simpler kernels presenting an intermediate step between the RPA (which can be viewed as TDDFT with only the Hartree kernel) and the BSE[13]. We illustrate the resulting Jacob's Ladder of optoelectronic properties in Fig. 1, where Hedin's equations stands for the full self-consistent MBPT as summarized in Ref. 14. In passing, we mention that the arguably most popular advanced electronic structure method, namely calculations within the so-called *GW* approximation, actually comprise a whole family of variants which differ regarding the degree of self-consistency of the Green's function denoted *G* and the screened interaction denoted *W*. We note that better scaling and/or more efficient variants of the described ab initio techniques to obtain optical properties exist, to name only a few: the reformulation into an initial-value problem[15], the use of the Lanczos algorithm[16], or the use of localized basis sets leading to sparse matrices[17]. Research in these directions remains a highly active field of research.

Machine learning (ML) offers a promising alternative or complement to the aforementioned traditional ab initio methods, and indeed working models have been reported by us and other groups that are able to predict frequency-dependent optical spectra of a material in fractions of a second[18–20]. Apart from being faster for relatively small cells, they also scale better, usually with *N* for message passing networks, where *N* is the number of atoms per primitive unit cell. All previous attempts at ML predictions of optical spectra have been trained on data calculated at the lowest reasonable theory level, i.e., IPA calculated based on DFT band structures. Therefore, the real-world value of these models remains limited because the inclusion of additional physical effects is critical to improve the agreement between the calculated spectra used as training data and experimentally measured spectra.

Training neural networks directly on high-level theoretical data from the upper rungs of either Jacob's Ladder is not currently feasible, because generating a sufficiently large dataset at this level of theory is prohibitively expensive. In machine-learning literature, such more accurate but more costly data is usually referred to as high-fidelity data. A promising alternative is transfer learning, where a model initially trained on a large set of low-level data is subsequently retrained on a much smaller set of high-level, i.e., high-fidelity data. This approach has been demonstrated in a materials science context in Ref. 21 for scalar quantities such as formation energy and band gap, and briefly in Ref. 19 for IPA spectra. In both studies, the models were first trained with data from GGA functionals and then refined using transfer learning with data from meta-GGA functionals. This corresponds to climbing up a rung on Perdew and Schmidt's original Jacob's Ladder (step (i) of spectra calculations), but staying on the IPA level regarding step (ii). In the special case of optical spectra, refinement from a GGA to, say, a meta-GGA functional typically results in a more or less constant shift of the spectra to higher energies.

An open question is whether transfer learning is similarly successful for more complex changes in optical properties, i.e., climbing the Jacob's ladder corresponding to step (ii) of the spectra calculation described above, such as the inclusion of local-field effects. These have a pronounced and much more complex impact on the optical spectra, redistributing spectral power and shifting peaks more or less significantly in a manner depending strongly on how similar the specific material is to the homogeneous electron gas[22]. The standard way to incorporate way these local field effects is by solving the RPA equations instead of using the IPA.

We briefly summarize the RPA equations formulated for a plane-wave basis set. Starting point is the so-called irreducible susceptibility $\chi^0(\mathbf{q}, \omega)$ at frequency $\omega$ and (crystal-) momentum transfer $\mathbf{q}$. For non-magnetic systems at zero temperature, it is given by[4]:

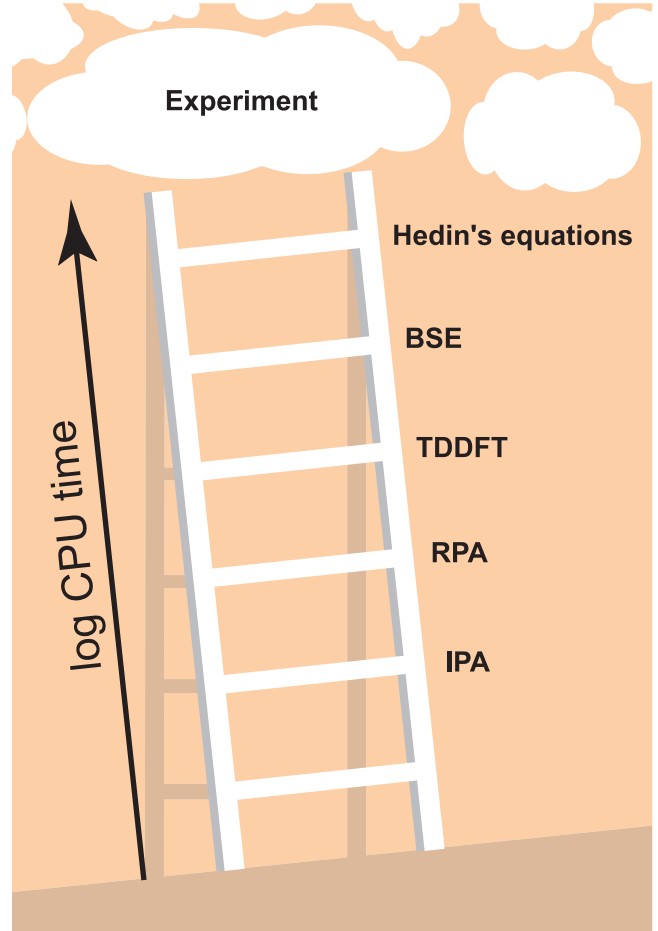

$$\chi^0_{\mathbf{GG'}}(\mathbf{q}, \omega) = 2 \sum_{vc} \int_{\mathrm{BZ}} \frac{d\mathbf{k}}{(2\pi)^3} \rho^*_{vc\mathbf{k}}(\mathbf{q}+\mathbf{G}) \rho_{vc\mathbf{k}}(\mathbf{q}+\mathbf{G}')$$
$$\sum_{\beta = \pm 1} \frac{\beta}{\omega + \beta(\epsilon_{v\mathbf{k}-\mathbf{q}} - \epsilon_{c\mathbf{k}} + i0^+)} \tag{1}$$

with the generalized dipole matrix element $\rho_{vc\mathbf{k}}(\mathbf{q}) = \langle \phi_{c\mathbf{k}}(\mathbf{r})|e^{i\mathbf{qr}}|\phi_{v\mathbf{k}-\mathbf{q}}(\mathbf{r})\rangle$. In almost all applications, Kohn-Sham eigenstates and eigenvalues are used as wave functions $\varphi_{n,\mathbf{k}}$ and energies $\epsilon_{n\mathbf{k}}$ (of valence and conduction bands, $n = \{v, c\}$, respectively, and $\mathbf{k}$ in the Brillouin Zone (BZ)). $\chi^0$ and all related matrices are large matrices with respect to the reciprocal lattice vectors $\mathbf{G}$.

This fact makes the RPA itself, i.e., the inversion of the Dyson equation for the full susceptibility $\chi$:

$$\chi_{\mathbf{GG'}}(\mathbf{q}, \omega) = \chi^0_{\mathbf{GG'}}(\mathbf{q}, \omega) + \sum_{\mathbf{G_1}} \chi^0_{\mathbf{GG_1}}(\mathbf{q}, \omega) \nu_{\mathbf{G_1}}(\mathbf{q}) \chi_{\mathbf{G_1G'}}(\mathbf{q}, \omega) \tag{2}$$

numerically expensive, since it involves matrix inversions and products of $\chi^0$, the Coulomb potential $\mathbf{v}$, and the desired quantity $\chi$. From the latter, the so-called microscopic dielectric function $\varepsilon^{-1}_{\mathbf{GG'}}$

$$\varepsilon^{-1}_{\mathbf{GG'}}(\mathbf{q}, \omega) = \delta_{\mathbf{GG'}} + \nu_{\mathbf{G}}(\mathbf{q}) \chi_{\mathbf{GG'}}(\mathbf{q}, \omega) \tag{3}$$

**Fig. 1 | Illustration of the proposed Jacob's Ladder of optoelectronic properties.** CPU time and accuracy generally increase significantly as one proceeds towards higher rungs, but the exact distance between rungs in terms of CPU time depends strongly on implementation, and the distance between rungs in terms of accuracy depends strongly on the material in question.

and finally the macroscopic dielectric function

$$\varepsilon(\omega) = \lim_{\mathbf{q} \to 0} \frac{1}{\varepsilon_{\mathbf{00}}^{-1}(\mathbf{q}, \omega)} \qquad (4)$$

are easily obtained. The matrix size, i.e., the number of $\mathbf{G}$-vectors included throughout a calculation, is an important parameter to be controlled for numerical convergence, see "Methods" section.

The IPA as used in the IPA-based OPTIMATE model[18] can be considered as a crude approximation to the RPA where only the first $\mathbf{G}$-vector, $\mathbf{G} = (0, 0, 0)$ is included in all matrix operations, which obviously strongly reduces the computational costs involved. The inclusion of local fields, i.e., finite $\mathbf{G} \neq 0$, generally leads to a more-or-less strong frequency-dependent reduction of $\mathrm{Im}(\varepsilon)$, but can also lead to peaks shifting, as the inclusion of the local field effects acts as an effective electron-hole interaction[22]. The differences between the RPA and IPA spectra are strongly material dependent, as we will see in the results below.

To answer how successful transfer learning is for more complex changes to the dielectric function, we create a database of optical properties calculated in the RPA and train graph attention networks (GATs)[23], specifically OPTIMATE models[18], on subsets of various size, either with or without previous training on IPA data. We find that transfer learning on relatively small datasets results in models that are able to predict spectra at higher levels of theory. Furthermore, we discover that transfer learning only on data from small cells leads to good generalization performance also for larger cells. As a side effect, we create and make openly available a model for optical properties at a higher theory level than previously published models, namely the RPA, see "Data availability" statement and "Code availability" section.

## Results

We have calculated RPA spectra for all materials in the IPA database[24] that have at most 8 atoms in the unit cell, i.e., a total of about 6000 spectra. The structures were originally taken from the Alexandria database of theoretically stable materials[25]. They include semiconducting and insulating compounds that contain only main group elements from the first five rows of the periodic table, have a distance from the convex hull less than 50 meV times the number of atoms in the unit cell, and a band gap greater than 500 meV. The data spans a wide range of crystalline systems, covering, e.g., common binary and ternary semiconductors and insulators, but also more exotic compounds such as layered two-dimensional systems and noble gas solids. A visualization of the elemental diversity of the data is given in Supplementary Fig. 3 and Supplementary Note 3. Additional computational details for the ab initio calculations are given in the "Methods" section. For the generated data, see the "Data availability" section.

For consistency, we employ the OPTIMATE architecture of GATs as in Ref. 18. The model is sketched in Supplementary Fig. 2 and takes as input a crystal structure and outputs a frequency-dependent optical property. An input crystal structure is first converted into a multigraph with the corresponding element encoded on each node and distances between atoms encoded on each edge. The model itself consists of a multilayer perceptron which acts on each node separately to process the elemental information, followed by three layers of message passing using the improved Graph Attention Operator of Ref. 23. Node-level information is then pooled using softmax vector attention to obtain a single high-dimensional vector which characterizes the material[18]. This vector is then passed through another multilayer perceptron to obtain the target optical property. The optical property itself is represented as a 2001-element vector, which corresponds to the optical property sampled in 10 meV steps from 0 eV to 20 eV. We refer to the original publication[18] for a more detailed description of its architecture.

In this work, we focus on models trained to predict the imaginary part of the trace of the frequency-dependent dielectric tensor $\varepsilon(\omega)$.

The imaginary part of the dielectric function determines together with the refractive index $n$ the absorption spectra $\alpha(\omega)$ via

$$\alpha(\omega) = \frac{\omega}{c} \frac{\mathrm{Im}(\varepsilon(\omega))}{\mathrm{Re}(n(\omega))} \qquad (5)$$

where in general $\alpha$, $n$ and $\varepsilon$ are all tensorial properties. The trace of these tensorial properties is an estimation of their polycrystalline average and allows for an easier handling of them[26]. We focus specifically on the dielectric function instead of, e.g., the refractive index or the absorption spectrum, as it is the direct result of the ab initio calculations discussed above, and is also usually reported in optical characterizations of materials, e.g., using ellipsometry.

We consider and compare two different strategies for learning the optical properties of materials at the RPA level, namely direct learning (DL) and transfer learning (TL). For DL, one initializes the weights of OPTIMATE randomly and trains directly on the expensive high-level optical spectra, i.e., the RPA spectra in our case. For TL, one trains OPTIMATE first on a large dataset of optical spectra of a lower theory level, i.e., in our case IPA, and then continues the training on a small number of optical spectra of a higher theory level, i.e., data calculated in the RPA. Since the RPA spectra in this work are calculated with a 300 meV broadening, we use the corresponding IPA-OPTIMATE model, i.e., the one trained on the trace of the imaginary part of the dielectric function under IPA, $\mathrm{Im}(\bar{\varepsilon}_{\mathrm{IPA}})$, with a 300 meV broadening. In this process, we retrain all learnable parameters of the base model[18] starting from the IPA weights, because, in contrast to many other applications involving, e.g., large language models, the additional cost is negligible, and recent publications[21,27] have shown that this is generally the best strategy for transfer learning graph neural networks for materials science, even when transfer learning between closely related properties. To avoid data leakage, we reuse the training, validation, and test sets used in the original training of OPTIMATE[18], leaving us with a total training set of 4610 materials and associated spectra, a validation set of 603 materials and a test set of 639 materials. This ensures that the transfer-learned OPTIMATE model has not seen any of the materials in the RPA validation and test set during its original training on IPA data. To compare the data efficiency of the DL and TL, we divided the training set into randomly formed subsets of 100, 300, 1000, 3000, and 4610, i.e., the entire training set. To ensure a fair comparison, we optimize both the architecture and the hyperparameters of the OPTIMATE models used for the DL strategy. The architecture of the OPTIMATE models used for the TL strategy is necessarily kept fixed, so only the hyperparameters of the optimizer are tuned. The training procedure, the selection of the final models, and their hyperparameters are described in more detail in the "Methods" section.

As error metric to compare two spectral properties $X(\omega)$ and $Y(\omega)$ in a scale-invariant way, we define the similarity coefficient (SC)[28]

$$\mathrm{SC}[X(\omega); Y(\omega)] = 1 - \frac{\int |X(\omega) - Y(\omega)| \, d\omega}{\int |Y(\omega)| \, d\omega} \qquad (6)$$

In addition, we use the mean square error (MSE)

$$\mathrm{MSE}[X(\omega); Y(\omega)] = \frac{1}{\omega_{\max}} \int |X(\omega) - Y(\omega)|^2 \, d\omega \qquad (7)$$

as another measure of similarity between spectra. During training, we use the $L_1$ loss, outside of machine learning commonly referred to as the mean absolute error (MAE):

$$\mathrm{MAE}[X(\omega); Y(\omega)] = \frac{1}{\omega_{\max}} \int |X(\omega) - Y(\omega)| \, d\omega \qquad (8)$$

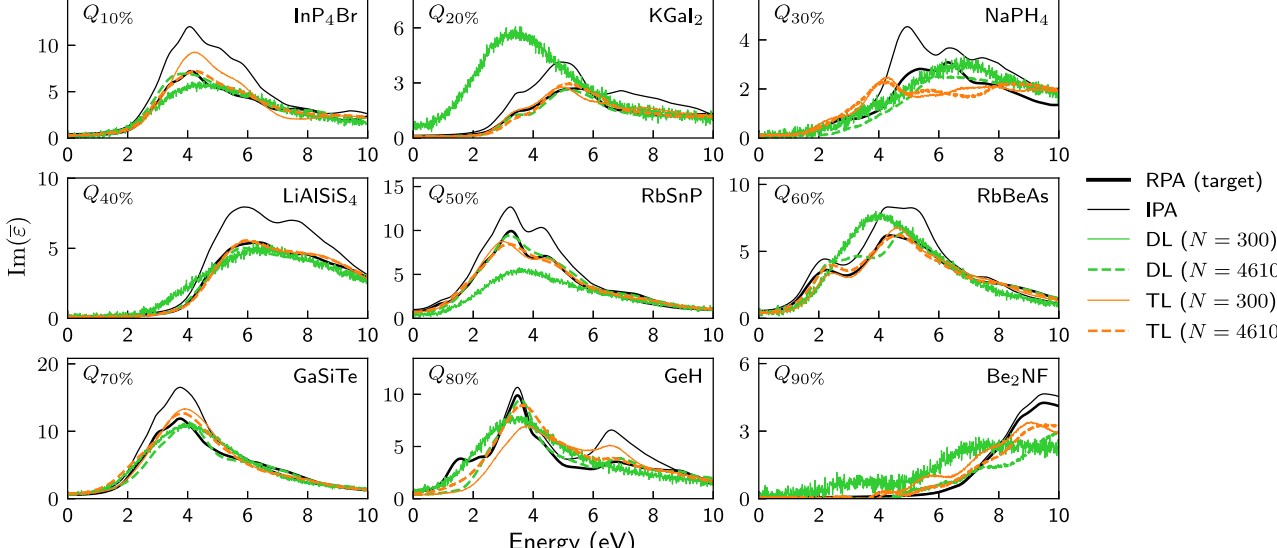

**Fig. 2 | Spectra as predicted via DL and TL after training on datasets of different sizes.** Spectra of nine materials selected at the 10% quantile, the 20% quantile, ..., the 80% quantile and the 90% quantile of the test set ordered according to the similarity coefficient SC[IPA$_{DFT}$; RPA$_{DFT}$]. All calculated or predicted spectra were obtained for a larger energy range of 0–20 eV, but are shown for a smaller energy range to emphasize the differences. All IPA spectra (thin black) differ considerably from the RPA spectra (thick black line). The latter are reasonably well predicted via transfer learning (orange) on large (thick dashed lines, 4610 materials) or even small (thin lines, 300 materials) trainings sets, whereas direct learning (green) on 300 materials yields unphysical, overly broad and very noisy spectra. Much larger training sets would be required for good and physically sensible results with direct learning. $N$ in the legend indicates how many materials were used for training.

We use a shorthand, e.g., SC[RPA$_{DFT}$; IPA$_{DFT}$] to refer to SC[$\bar{\varepsilon}(\omega)^{RPA}_{DFT}$; $\bar{\varepsilon}(\omega)^{IPA}_{DFT}$], i.e., in this example, the SC between $\bar{\varepsilon}$ obtained via DFT using the RPA in reference to $\bar{\varepsilon}$ obtained via DFT using the IPA.

**Comparison of direct learning and transfer learning**
The main results of this work are summarized in Figs. 2 and 3, which compare both learning strategies for various training-set sizes. The examples in Fig. 2 show that DL on a small dataset of 300 materials leads to noisy, unphysical spectra consisting mostly of a single broad peak (which, perhaps surprisingly, is often more or less correctly positioned). In contrast, the TL strategy results even for small trainings sets already in realistic spectra, showing quite good quantitative agreement. For larger training set sizes, e.g., for 4610 training materials as shown, both strategies lead to a similar prediction accuracy. This is shown in more detail in Fig. 3, where the median SC, Eq. (6), and the median MSE, Eq. (7), of the test set for various training set sizes are compared between both strategies. Both error measures are similar for both strategies when the training set is large enough, but the transfer learning strategy already achieves similar error measurements with just 300 materials as the direct learning strategy with about 3000 materials. All models perform better than the possible baseline of just using the IPA spectra. Numerical values for the IPA baseline, i.e., $G_{max} = 0$, as well as for the non-converged $G_{max} = 2000$ mRy are given in Supplementary Table 4. Parity plots for both strategies trained on 300 and 4610 materials are shown in Supplementary Fig. 4 and Supplementary Note 4 and yield similar results.

In short, transfer learning allows accurate quantitative prediction of spectra with a training set of only the order of 100–1000 materials. Furthermore, we believe that the observed convergence in model performance between both strategies is likely due to the RPA training set approaching the size of the initial IPA training set for the model, as it is often observed in TL that as the underlying training set increases, TL accuracies also improve[21].

Hitherto, we have seen that a few (about 300) numerically more expensive RPA calculations are sufficient to retrain the OPTIMATE IPA model. Now, we turn to the question of whether we can pick particularly cheap examples for the TL strategy. So far, the data used for transfer learning consisted of random subsets of the data used to train the underlying model. As discussed in the introduction, almost all computational techniques scale unfavorably with the system size (often $\mathcal{O}(N^3)$ or worse), as opposed to graph neural networks, which generally scale as $\mathcal{O}(N)$, with $N$ being the number of atoms in the unit cell. If instead of using a random subset of the original training data one could instead only use data from small unit cells, one might significantly reduce computational time. We therefore transfer learn the IPA model on a subset of the training data consisting of only the materials with at most 2, 3, 4, 5 and 6 atoms per primitive unit cell (using the optimal hyperparameters for a training set size of 1000). The results are shown as blue lines in Fig. 3 and they are impressive: Even with TL on materials which only contain up to 2 atoms per primitive unit cell, which are only around 60 materials in total, a median SC of 0.8 is achieved. With only up to 4 atoms per primitive unit cell, i.e., around 1500 materials, median MSEs below 0.1 and SC above 0.85 are achieved.

One could speculate that learning on small cells primarily improves the performance on other small cells, while not improving the errors on larger cells. But that is not the case: Fig. 3 (bottom) shows the SC for the materials in the test set grouped according the number of atoms per primitive unit cell, evaluated for the model trained on cells with up to 4 atoms per primitive unit cell. The error for all unit cell sizes is equally small, i.e., even when transfer learning only on materials with small cells, the properties of materials with larger cells can be predicted equally as well. If this were a general result, it would be very encouraging for related future work, e.g., climbing further rungs towards accurate prediction of experimental spectra.

**Deeper investigation into transfer learning**
We further investigate the success of transfer learning. Figure 4 quantifies the observations that are suggested by the illustrative examples of Figs. 2 and 3. We start with the one-dimensional histograms marking rows and columns. The most important observation is that the TL-RPA predicts the ab initio DFT-RPA (see panel b) about as well as the underlying ML-IPA model[18] predicts the DFT-IPA (panel d).

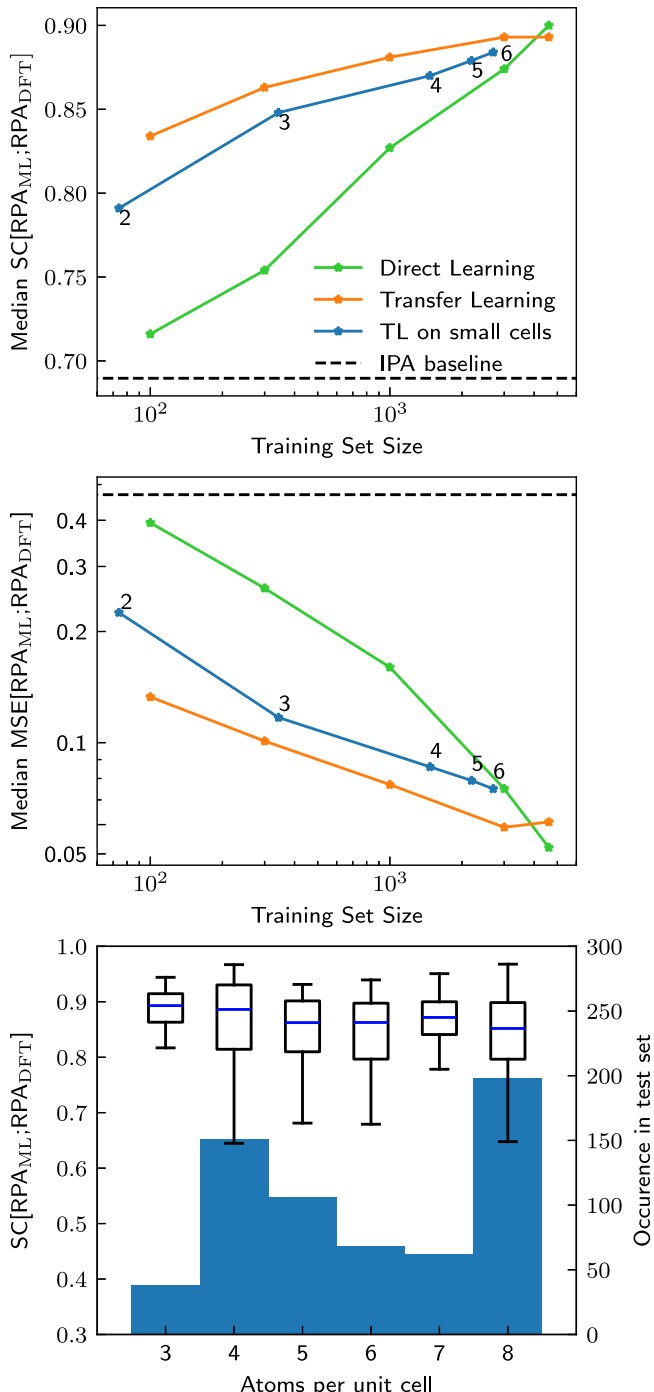

**Fig. 3 | Comparison between DL and TL for different training set sizes.** Median SC (top) and median MSE (middle) on the test set for direct learning (green) and transfer learning on subsets of different size of the training set (orange) or exclusively on systems with small unit cells (blue) as a function of the training-set size (indicated by the stars). Using the DFT-calculated IPA spectra as a proxy for the RPA spectra (taken over the entire test set) is given as a baseline in the black dashed line. The small number next to each data point on the blue curve indicates the maximum number of atoms per primitive unit cell. Bottom: Boxplot of the distribution of $SC[RPA_{ML}; RPA_{DFT}]$ on the full test set after transfer learning only on materials with up to 4 atoms per primitive unit cell. The test set is split up by number of atoms per primitive unit cell. The blue center line of the boxplot shows the median, the boxes show the interquartile range, and the whiskers extend out from the box to the furthest point lying within 1.5 times the interquartile range. As can be seen, transfer learning only on materials with few atoms per unit cell generalizes to materials with more atoms per unit cell. The number of materials in the test set with a given number of atoms per primitive cell is given by the background histogram. No materials with 1 and only very few materials with 2 atoms per primitive unit cell are present in the test set.

whether the models can predict the RPA spectra and whether another model can predict the IPA spectra. Most peculiarly, the numerical values of the corresponding SCs are almost the same when transfer learning already on small datasets (panel bd). This means that the RPA spectra for a given material can be predicted just as well as the IPA spectra. This correlation is made even more clearer when training on 4610 materials, and the same agreement is recovered for DL on such dataset sizes, see Supplementary Fig. 1. One might have expected—wrongly, as it turns out—that the RPA spectra would be more difficult to predict than the IPA spectra, since they include additional physical effects.

From a machine-learning point-of-view, the fact that RPA and IPA spectra are equally well predictable might not be surprising, by—perhaps overly reductionistly—assuming the models treat the RPA spectra compared to the IPA spectra as just another curve defined in a learned materials space. As we have shown recently, the models construct expressive internal representations of the material space[29]. Therefore, we propose that the models reconstruct very general features of the material space (similarity between materials in general) contained in the training set before converting them to a specific property (in this case the RPA dielectric function).

To investigate this further, we compare in Fig. 5 how well each material is predicted by both strategies for various training set sizes. Again, TL performs much better than the DL strategy for the small training set. Both strategies perform similarly well for the large training set, where an almost linear correlation exists, with possibly a slight advantage for DL. Thus, not only is the predictability between the IPA spectra and the RPA spectra strongly correlated, but the predictability of the RPA spectra for a given material is also approximately independent of the training strategy employed (once the training set is large enough).

## Learning the similarity

Finally, we investigate whether it is possible to directly predict the similarity between the DFT-IPA spectra and the DFT-RPA spectra. This could be useful, e.g., when deciding which level of theory is necessary to model a given material. For this purpose, we train a model with the same architecture as OPTIMATE[18], except that we change the output dimension of the final output layer from a dimension of 2001 to 1, i.e., the dimension of the SC (see Supplementary Note 5 for training details). The results are shown in Fig. 6. As can be seen, the similarity coefficient between the IPA and the RPA spectra can be predicted extremely well, with a mean and median absolute error of 0.026 and 0.018, respectively. This bodes well for further studies. One might, e.g., design a high-throughput workflow which incorporates such models to effectively select the highest level of theory necessary to describe the properties for a given material, saving valuable computational resources.

Even the poorly trained DL prediction (a) is not worse than the possible baseline of simply using the DFT-calculated IPA spectrum (c), see also Fig. 3.

The first row of the two-dimensional histograms in Fig. 4 addresses the question of whether the quality of ML prediction on the RPA level depends on how much the ab initio RPA and IPA spectra actually differ, in other words, whether the ML predictions are worse for difficult cases with small $SC[IPA_{DFT};RPA_{DFT}]$. Obviously, there is no (strong) correlation, the models predict RPA spectra more or less well, regardless of how much the RPA changes the IPA spectra. While this may be obvious for the direct learning strategy (panel ac), this is far from obvious for the transfer learning strategy (panel bc), which starts training from a model which predicts the IPA spectra.

Examining the bottom row of two-dimensional histograms in Fig. 4 reveals an even more interesting fact: There is a correlation between

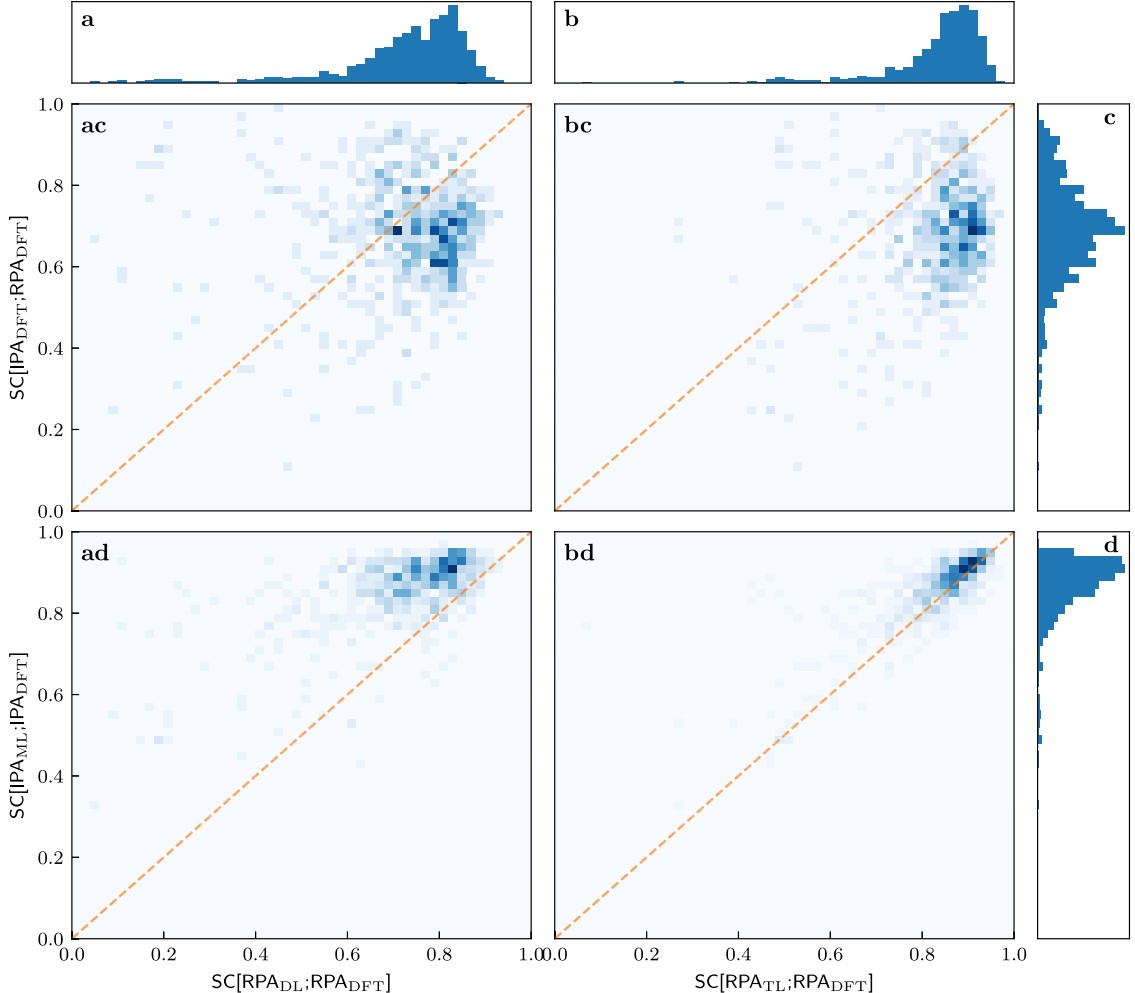

**Fig. 4 | Correlations between various similarity coefficients on the test set after training on 300 materials.** The columns refer to SC[RPA$_{DL}$;RPA$_{DFT}$] **a** and SC[RPA$_{TL}$;RPA$_{DFT}$] **b**, while the rows refer to SC[IPA$_{DFT}$;RPA$_{DFT}$] **c**, SC[IPA$_{ML}$;IPA$_{DFT}$] **d**. The histogram in the first row shows that the ab initio IPA spectra are only a rough approximation of the ab initio RPA spectra. The histogram in the second row reproduces the observation of Ref. 18 that very good ML predictions of the DFT-IPA spectra are possible. The histograms in the top panels show that TL predicts the ab initio RPA spectra much better than DL. The two-dimensional histograms in the center show the correlations between the SCs defining columns and rows. All SCs are evaluated on the test set. The dashed orange line marks the angle bisector.

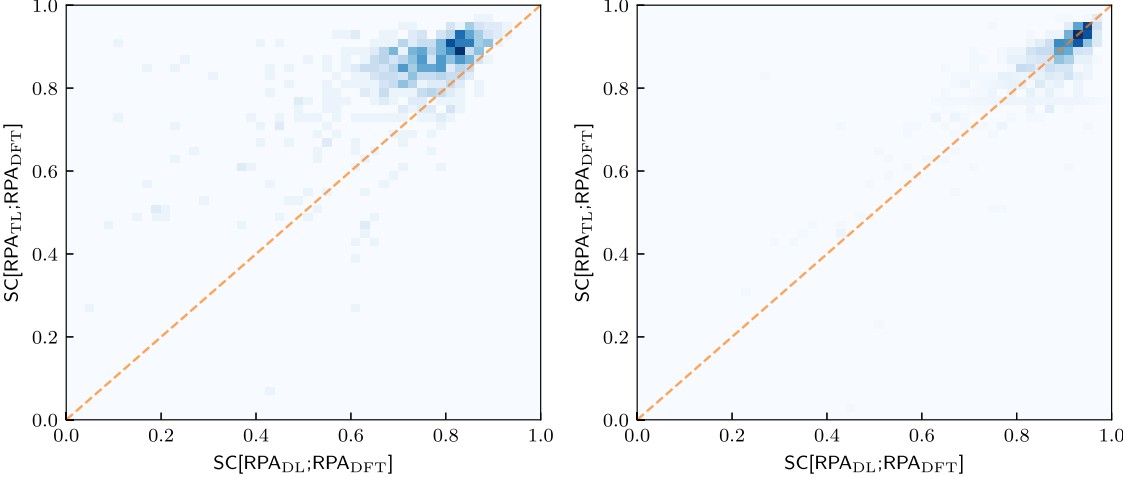

**Fig. 5 | Comparison between TL and DL for different training set sizes.** 2D-Histograms comparing the similarity coefficients SC[RPA$_S$; RPA$_{DFT}$], $S$ = DL, TL, on the test set for both training strategies after training on 300 (left) and 4610 (right) materials. The orange line marks the angle bisector.

## Which materials are well described by the IPA already?

The database presented in this work is the largest database of well-converged RPA calculations that we are aware of. Apart from being of direct practical use, one might hope to glean physical insight from such a database, for example by asking "Which (classes of) materials are well described by the IPA already", or framed in another way, "Which materials possess very strong local-field effects". From a theoretical point of view, the IPA is in essence a maximal truncation of a basis set (see Eq. (2)), and treats the susceptibility as being constant over the entire unit cell[3,22]. How correct this is obviously varies from material to material, and one might attempt to correlate it with the anisotropy of the electron density, for example. We instead make use of our descriptor $SC[RPA_{DFT};IPA_{DFT}]$ for an initial attempt to answer this question. Manually inspecting the materials with the lowest similarity $SC[RPA_{DFT};IPA_{DFT}]$ (see "Code availability" statement) reveals that most of these materials are ones that either consist of condensed weakly bound clusters or molecules (e.g., $S_8$-rings, $N_2O$, $CO_2$, $N_2$) or materials including many halogen or nitrogen atoms. The response of the former class is spatially inhomogeneous more or less by definition. Intuitively, strong local-field effects of the latter class might be explained by its very electronegative elements, which lead to unequal charge distributions and anisotropic screening. However, an explanation of low $SC[RPA_{DFT};IPA_{DFT}]$ values based on electronegativity alone suggests small SC values for oxides as well, which is not clearly supported by the data. Direct inspection of the materials with the largest $SC[RPA_{DFT};IPA_{DFT}]$ did not show any obvious material clusters.

We therefore proceed with an alternative approach. We extract the latent activations for the $SC[RPA_{DFT};IPA_{DFT}]$-prediction network after the pooling step and reduce them to two dimensions using the UMAP algorithm[30] (for details, see "Methods" section). As we have previously shown, a UMAP after the pooling step (on our specific architecture) generally clusters according to chemical principles and can be interpreted as a map of the chemical space from the point of view of the trained model[29]. The resulting UMAP is shown in Fig. 7, and one can clearly see that the materials with extremely low similarity cluster (materials with many halogen atoms to the right, azides, nitrates etc. to the lower left), while materials with average or high similarity are more spread out. The identity of each dot (i.e., each material) is identified via an interactive version of this UMAP with which one can investigate where each material is placed (see "Data availability" section).

## Discussion

The overall advantages of accelerating the prediction of optical properties are obvious: Besides allowing for the screening of large material databases, they also allow for the prediction and analysis of the optical properties of previously prohibitively complex systems, e.g., defects, nanoparticles, biomolecules, and large and dynamic systems like liquids simulated via molecular dynamics. The value of this work in regards to that goal is twofold: From a materials scientist's perspective, we provide a database of converged RPA calculations and OPTIMATE models to predict the RPA spectra of materials in a fraction of the time that an actual RPA calculation would take. As can be seen from the computational time spent (see "Methods" section), creating such a model requires a significant initial investment. However, processing new materials through a machine learning model is essentially free. As a side effect, the generated data illustrates that the RPA and the inclusion of local-field effects significantly change the optical spectra of most materials. We make the RPA database freely available and estimate that many more physical insights can be gained from a more detailed analysis of it.

From a machine learning perspective, we present an impressive example that transfer learning of graph networks works extremely well

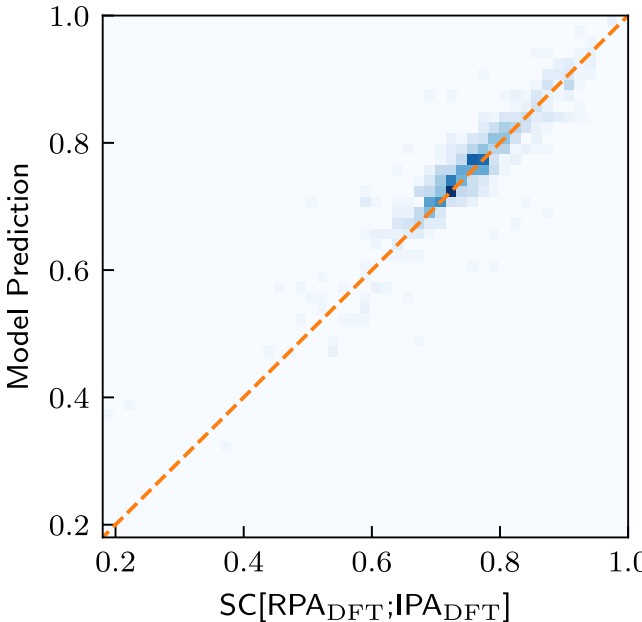

**Fig. 6 | Predicting the similarity between the RPA and the IPA.** Prediction of $SC[RPA_{DFT}; IPA_{DFT}]$ on the test set after training on the entire training set. The straight line marks the angle bisector.

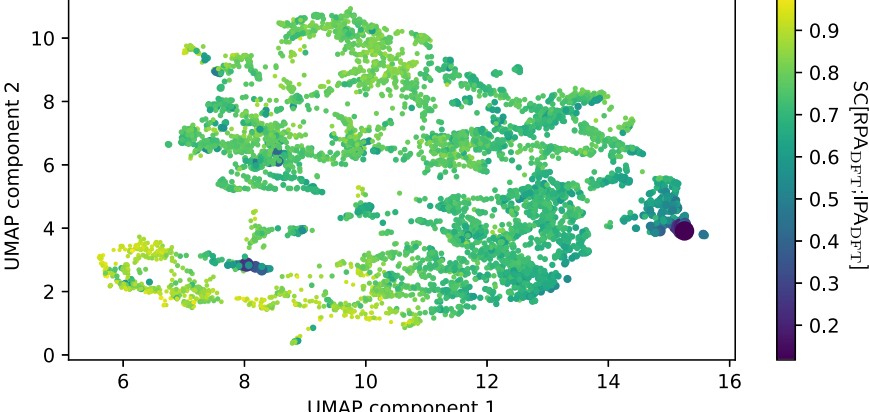

**Fig. 7 | UMAP of latent embeddings of SC-prediction model, colored by true SC.** Each dot represents a material. The color and size of each dot are related to the SC, with larger/darker dots showing less similarity between RPA and IPA. There are two distinct clusters of very low similarity. The right purple cluster consists of compounds with extreme amounts of halogens (e.g., $KF_3$, $RbBrF_6$, $XeF_2$, etc.), while the purple cluster closer to the lower left corner consists mostly of azides, nitrates and other nitrogen-containing compounds (e.g., $NaN_3$, $LiNO_3$, $N_2$, etc.).

for spectral properties, and that this holds true even with very limited high-fidelity data, reducing the necessary up-front cost of generating training data immensely. This agrees with similar previous observations for scalar properties[21]. We believe that this is a viable path to eventually achieving spectra prediction which agree with experimental data and, thus, allow actual concrete predictions for novel optical materials, something which so far has been completely out of reach. This can be achieved, e.g., by transfer learning on $GW$-BSE spectra. It remains to be seen whether transfer learning directly to the highest level of theory, e.g., directly from PBE-IPA to $GW$-BSE, will be more or less effective than taking intermediate steps, e.g., first transfer learning from PBE-IPA to PBE-RPA and then, in a second transfer learning step to $GW$-BSE. For this route, the observations that one can use cheap calculations for the TL strategy and that graph attention networks can be used to predict how many rungs of Jacob's ladder must be climbed, are particularly noteworthy and promising.

## Methods

### Ab initio calculations

We use QUANTUM ESPRESSO[31,32] for the ground-state calculations and YAMBO for the calculation of the RPA spectra. The DFT calculations were performed using PBE[33] as exchange-correlation functional and optimized norm-conserving Vanderbilt pseudopotentials from the SG15 library (version 1.2)[34]. First, the plane-wave cutoff (initial value of 60 Ry, step size of 5 Ry) and the **k**-point grid (initial value of $n_{\mathbf{k}} = 1500$ per atom, increased in even subdivisions) of the self-consistent ground-state DFT calculations were converged in regards to the total energy, with a convergence threshold of 1 kcal/mol per atom in the primitive unit cell. Afterwards, spectra are calculated on 2001 equally spaced points between 0 and 20 eV using a fixed k-point grid, defined through a structure-independent reciprocal density of $n_{\mathbf{k}} = 1500$ per atom as defined in PYMATGEN, which was shifted off symmetry[35]. The number of **G**-vectors is converged (separately for each direction of the electric field) in steps of 2000 mRy until a SC (Eq. (6)) of above 0.9 is reached. The vast majority of materials reach convergence at 4000 mRy. It is reasonable to expect that a smaller step size would have lead to convergence earlier. The step size was chosen based on our previous work on converging $G_0W_0$ calculations, where convergence is reached at much larger values[36]. The reason behind this earlier convergence is not immediately clear to us. The mean SC between the penultimate and the final calculation is around 0.97 (mean taken over all materials, and over all directions). The spectra are calculated for a spectral broadening of 300 meV. In total, the ab initio calculations used 193,639 CPU hours on the MaPacc4 high-performance cluster of TU Ilmenau. The distribution of CPU hours per material is long-tailed, with the mean per material being 26.2 h, and the median being 7.8 h.

### Machine learning

The optimal parameters for each strategy and dataset size were found separately. For the direct learning strategy, the optimal network architecture for predicting Im($\varepsilon$) under the IPA with a broadening of 300 meV from Ref. [18] is used for an initial scan of the optimizer hyperparameters. These hyperparameters were then used for a broad search over possible architectures. The optimizer hyperparameters were then scanned again for the optimized architecture. Each scan was performed using 5-fold cross-validation on the corresponding subset of the training data. The resulting models were then evaluated on the validation set to select the ideal parameters. The final optimized parameters are available in the SI, see Supplementary Note 2 and Supplementary Tables 1 and 2. Finally, separate models were trained on the full respective subset, i.e., without cross-validation, and evaluated on the test set to obtain the results shown in this paper. For the transfer learning strategy, only the optimizer hyperparameters were optimized, in the same manner as described above, since the architecture necessarily had to be kept fixed. All models are trained using an $L_1$ loss and the Adam optimizer[37]. Training and test set errors are given in Supplementary Table 3. Training a machine learning model on the full training data set takes between 5 to 10 min (tested on RTX 3060 mobile, RTX 3080, RTX 5090, and a A100). A full forward pass on a randomly selected material (agm004850436, $Ca_4SbTe_2Br$) using a trained model takes a total of 14.8 ms on a Laptop with a RTX 3060 mobile (timing being done using the timeit functionality of JUPYTER), taking into account reading in the CIF file as a PYMATGEN structure object ($6.7 \pm 0.1$ ms), converting the structure into a graph ($4.6 \pm 0.1$ ms), and running the forward pass ($3.5 \pm 0.1$ ms).

The UMAP in Fig. 6 is generated using a number-of-neighbors parameter of 30 and a minimum distance of 0.1. As in Ref. 29, we find that a few materials in the test set are incorrectly placed when using the latent embeddings of the SC-prediction model. We therefore retrain the SC-prediction model on the entire database and use its latent embeddings after the pooling step for creating the UMAP. As models running on GPUs are ever so slightly nondeterministic and the UMAP algorithm is nonlinear, we extract the latent embeddings once and store them, so that the UMAP can be reproduced. In practice, we find visually similar but arbitrarily rotated UMAPs if we do not do this.

## Data availability
The RPA data and the weights for the optimized models generated in this study have been publicly deposited in the Figshare database under accession code[38]. This includes structures, spectra, bandgaps and computational parameters. In addition, we also supply raw input and output files of our calculations at Ref. 38. This also includes, e.g., charge densities and spectra calculated with intermediate values of $G_{max}$, and an interactive version of the UMAP shown in Fig. 7.

## Code availability
The used third-party codes YAMBO and QUANTUM ESPRESSO are available at the time of publication of this work at https://www.yambo-code.eu and https://www.quantum-espresso.org, respectively. The workflows used to generate the RPA data presented here as well as scripts to use the models and to carry out the analysis done in this work are available at Refs. 39,40. Possible future updates are available at Refs. 41,42.

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

## Acknowledgements

We thank the staff of the Compute Center of the Technische Universität Ilmenau and especially Mr. Henning Schwanbeck for providing an excellent research environment. This work is supported by the Deutsche Forschungsgemeinschaft DFG (Project 537033066, awarded to E.R).

## Author contributions

M. Grunert and M. Großmann conceived the idea; all authors designed the study; M. Grunert wrote a workflow for high-throughput RPA calculations and performed them; M. Grunert trained all machine learning models; M. Grunert and M. Großmann analyzed the data, M. Großmann assisted in visualizing the results; all authors modified and approved the manuscript; E.R. supervised the work and the entire project.

## Funding

## Competing interests

The authors declare no competing interests.
