## [Peer review file · Nature Communications]

Machine learning climbs the Jacob's Ladder of optoelectronic properties

Corresponding Author: Mr Malte Grunert

Version 0:

Reviewer comments:

Reviewer #1

(Remarks to the Author)

Review for "Machine learning climbs the Jacob's ladder of optoelectronic properties"

The authors extend their GAT-ML scheme for optical spectra (Ref. 12 in the manuscript), by moving from GGA-IPA to GGA-RPA. To improve the efficiency of the ML training, the authors explore a transfer learning approach. Convincing evidence is presented that transfer learning from IPA to RPA outperforms direct learning of RPA spectra. Furthermore, the authors demonstrate that their ML scheme can predict optical spectra of larger systems not included in the training set. Finally, the authors address the "learnability" of RPA spectra, and try to predict the difference IPA/RPA through an additional ML scheme. Overall, the study is well sourced and conducted on a high technical level. The introduction is well written and clearly outlines the need for ML of RPA spectra. The results are clearly presented, and the conclusions drawn are logically sound. To further improve the quality of the manuscript, I propose the following points:

Major points:

A1) The authors present a large database containing thousands of RPA spectra for bulk solids. To the best of my knowledge, no such database exists so far, and thus these calculations alone constitute a significant achievement. Given that the inclusion of local-field effects in the RPA is the main advancement of this work, more effort should be spent on postprocessing and analysing the RPA data. For instance, the importance of local-field effects [as measured by $SC(IPA,RPA)$] can be analysed as a function of crystal structure and correlated to properties of the constituent elements. Here, the authors could make good use of their descriptor presented in Fig. 5, and attempt to reverse-engineer the atomistic origins of strong local-field effects. To support their analysis, the authors could also present a plot similar to their Fig. 1, demonstrating weak, medium, and strong local-field effects. Further, the importance of local-field effects can be correlated to scalar dielectric properties following for example Ibrahim and Ataca (see Fig. 5 in Ref. 13 of the present manuscript). Finally, one can go beyond analysing $SC(IPA,RPA)$ by looking at differences in the position and heights of the main peaks, or by resolving $SC(IPA,RPA)$ to different spectral ranges (IR, visible, UV).

A2) Figure 3: The authors demonstrate that ML accuracy surpasses the minimum baseline of the IPA itself. Given that the IPA can be interpreted as maximally truncating the basis set (lines 70-72), I suggest a more stringent control: RPA spectra with reduced basis set size. The accuracy of the RPA spectra can be plotted as a function of basis set size, and this convergence plot can be compared to the learning curves given in Fig. 2. The method section states that the RPA spectra are obtained iteratively starting with smaller basis sets (lines 210-211). Thus, the data points for the convergence plot should be readily available. In this way, the compute cost of the ML approach (creating the ML database and training the models) can be contrasted with the computational overhead of simply calculating the RPA spectra. Perhaps a concrete case can be made for the larger structures in the test set (7-8 sites in the unit cell, see Fig. 2).

Minor points:

B1) Introduction: machine learning should be discussed in comparison to other strategies aiming to reduce the computational cost of TDDFT/BSE. Such strategies encompass real-time propagation (see e.g. doi: 10.1103/PhysRevB.67.085307), the use of the Lanczos algorithm (see e.g. doi: 10.1103/PhysRevLett.96.113001), and basis set reduction (see point A2). The scaling of these approaches can also be contrasted with ML (see lines 133-136).

B2) Figure 1: The names of the 9 materials should be given in the plots. Is there any explanation why some spectra are easier/harder to predict for ML (see also point A1)?

B3) p. 4: The authors discuss SC and MSE as error metrics (Eqs. 5 and 6). In their previous work (Ref. 12 in the manuscript, SI), the authors indicate that a L1 loss is used for training. It should be clarified, whether this is still the case (Supplementary

Table 1 suggest an L2 loss is used). If L1 is still used, L1 losses should be defined on p.4 and L1 losses should be given as addition to errors shown in Fig. 2 (can be added as supplementary). Further, training set errors should be given as supplementary information, too.

B4) Figure 2: The authors should indicate IPA as baselines in the learning curves. If this overwhelms the plots, mean IPA errors can also be stated in the text (line 155).

B5) Figure 2: For very large training sets, direct learning seems to outperform transfer learning slightly. This is also discussed by the authors in lines 176-179. However, the authors discuss that SC of 0.9 represents the numerical accuracy of their spectra (line 211). Thus, it is conceivable that DL overfits and learns numerical noise in the training data. This could be tested explicitly (i) by calculating higher accuracy spectra, or (ii) by training DL and TL on lower accuracy spectra.

B6) Lines 162-175: The authors discuss the puzzling finding that RPA is as easy to learn as IPA. This could be due to the fact that the database does contain neither transition metals nor low-dimensional materials, where local-field effects would be more pronounced. In future work, it will be interesting to see, whether ML of BSE spectra is equally as easy. In truly non-local cases such as long-range charge-transfer excitons, the current ML scheme seems bound to fail due to the real-space cutoff in the model.

B7) Abstract: There is a grammatical error in the first sentence of the abstract, "approximation" is erroneously written twice. Title: "Jacob's Ladder" should be capitalized in the title. Line 185: the authors train a ML model, not a material.

Optional points:

C1) p. 1: The authors discuss RPA as the second step on the Jacob's ladder of optoelectronic properties. To convey the importance of their work to a broader audience, the authors could compare IPA, RPA, BSE spectra, and experimental spectra for a select number of materials. Then, the similarity coefficient can be used to contrast SC(IPA,RPA), SC(RPA,BSE), and SC(BSE,expt) (i.e. how big are the steps on the ladder). I recognize that BSE calculations are expensive, but perhaps the authors can take BSE spectra from the literature. Moreover, the presentation of the work might benefit from a small comic illustrating this Jacob's Ladder of optoelectronic properties (similar to the original one, see Ref. 1 in the manuscript).

C2) Fig. 4 contains somewhat redundant information and could be omitted (cf. Figs. 2 and 3 and Supplementary Fig. 1).

C3) lines 198-202: The authors discuss that their transfer learning scheme could be applied to enable the prediction of GW-BSE spectra. Perhaps it is beneficial to introduce an intermediate step that promotes GGA-IPA spectra to GW-IPA spectra. In this context, for instance the recent work of Knøsgaard and Tyghesen (doi: 10.1038/s41467-022-28122-0) could be useful. As an alternative to GW-IPA, meta-GGA could also be viable (e.g. Ibrahim and Ataca, Ref. 13 in the manuscript).

C4) Lines 183-190: The authors discuss that their similarity descriptor (IPA/RPA) could be used for high-throughput studies. Once future work has extended the current scheme to BSE, an interesting prospect is to adapt the similarity predictor for ML of exciton binding energies.

C5) Outlook: The prospect of combining ML interatomic potentials with ML optical spectra seems exciting. This could enable fast prediction of optical spectra of large and dynamic systems (e.g. liquids).

In conclusion, the study advances the emerging field of ML optical spectra through two main achievements: (i) The construction of a large database of RPA spectra, and (ii) the ML of RPA spectra with a transfer learning approach. In my opinion, not enough emphasis is yet given to achievement (i). The database presents a unique opportunity to explore the physics of local-field effects (see point A1). The authors have also indicated that the database will be publicly released. This will be highly useful for other researchers, and it will contribute to the long-lasting impact of this work. Transfer learning per se [achievement (ii)] is not as novel, but the authors give due credit to prior work. Still, the results are very encouraging. In point A2, I suggest a more challenging control for the ML approach. Such analysis could help to convince potential users to adopt the present scheme. Overall, the present study presents an important step towards ML of GW-BSE spectra, and I support publication in nature communications once the comments have been addressed.

Stefan Riemelmoser, EPFL

(Remarks on code availability)

Reviewer #2

(Remarks to the Author)

The paper reports the results of the fine tuning of a machine learning model predicting the optical spectra of a large dataset of crystals. The fine tuning involves going from a model trained on the independent particle approximation (IPA) to the results of a random particle approximation (RPA) approach, which in general provides better accuracy when comparing the computed spectra with experiment. Multiple experiments including the comparison of direct learning (DL) vs transfer learning (TL) are conducted to show and compare the performance of their model.

In general, the manuscript is very well written and explains the necessary theory and terms used nicely. The conclusions are clear and well evidenced. The experiments are well designed. The results are noteworthy as they demonstrate how little data of higher level theory for optical spectra is necessary to obtain a model that produces a similar high level of theory, even though similarly little data was used in other transfer learning tasks e.g. for when training machine learning interatomic potentials.

However, please address the following points for clarity:

For me, the target of the machine learning model is not quite clear. It seems to be the trace of the imaginary part of the dielectric function (as plotted in Fig. 1) or is it the whole imaginary part of the dielectric function as stated in the machine learning methods section? I could not find where this is explained further. Is the real part of the dielectric function also computed and if so, how is the coupling between the imaginary and the real part handled?

Could you provide in the SI the elemental compositions of your dataset? The diversity of the dataset is not clear to me. How many elements are involved? Are these only crystalline or also amorphous structures? High fidelity is claimed but no exact description is given.

Figure 1: what does the small dataset in the caption refer to? Is this the cell sizes or the dataset size? Also, a legend to easier compare the data would be nice. Also, what role does the scaling in y-direction of the spectra play here? When scaled the same, the IPA and RPA spectra for Q10% would be basically the same. Does it make sense to compare the absolute values of the spectra and not scale them to have the same integral first?

It would also be nice if the chemical structure of the calculated materials would be given, so a bit more information on the dataset would be obtained. Also, what are the proportions of the different quantiles in the dataset (just to estimate their influence on the training/testing)?

Please explain what the trace of the imaginary part of the dielectric function under IPA relates to, is this a simple adsorption spectrum?

Page 4: 'consisting of only the materials with at most 2, 3, 4, 5 and 6 sites per unit cell'. What are sites? Unique crystal sites or elements or elements on a specific crystal site?

Is the transfer learning on small cells evaluated on the full test set with also more sites in the cell? How large were the test sets?

Figure 3: why is it plotted? It does not make sense to compare the direct model with the similarity of the IPA and the RPA spectra.

Learning the similarity: The motivation is good, however why are no values given for which kind of material classes are adequately computed with IPA and which require RPA? Are there trends visible? This information would add to the scientific message of the paper.

(Remarks on code availability)

Reviewer #3

(Remarks to the Author)

The authors present a framework for predicting the optical spectra of semiconductors using graph neural networks and transfer learning from the lower-fidelity IPA calculations to the higher-fidelity RPA calculations. It is shown that while there are obvious differences between IPA and RPA estimates, transfer learning does help improve the ML prediction of the RPA-level optical spectrum. Analysis also shows that models can be trained mostly on smaller system sizes and extended to larger systems. The DFT dataset made available in this work will be very valuable.

I believe the work is strong, but a number of things must be clarified and changed before the manuscript is accepted for publication.

1. There is reference to the earlier publication that reports the DFT calculations and dataset, but I believe it would be helpful if the authors added some sentences about the chemical composition of the dataset. What kind of compounds are being used here? Does this contain oxides, halides, chalcogenides, etc., or is it restricted to particular classes of compounds? What role does chemistry play in this work?

2. It is also not clear to me how the GNN models are being trained- the crystal structure should be the input, and is the output the entire dielectric function spectrum? Is it broken into bins for prediction?

3. What specific GNN algorithms were used in this work, and which method works best?

4. Figure 1 is slightly confusing without a legend- I think something should be added to the right side of the figure to show what each color and type of line represent.

5. Related question on Figure 1- why is the solid green line alone very squiggly compared to the other lines?

6. It would also be useful to add actual parity plots showing ML prediction vs DFT in the SI.

7. Can the authors present a time (or computational expense) comparison between full DFT and the TL-based prediction?

(Remarks on code availability)

The code has not been made available yet- I believe it should be, even in the review stage.

Version 1:

Reviewer comments:

Reviewer #1

(Remarks to the Author)

The authors have addressed most of the points raised in an extensive and satisfactory manner. In particular, the additional analysis on the physics of the RPA is interesting (new Figure 7, see author response to point A1). I also welcome the decision of the authors to add supporting data to their publicly available database. A few minor issues still warrant further response, however:

Ad A2) Supplementary Table IV: I appreciate the clarifications concerning computational cost brought here and in the main text. The cutoff chosen by the authors ($G_{\max} = 2000$ mHa) is indeed an upper limit, given that no ML models actually reach $SC=0.97$. If the data is available, 2-3 more points at intermediate cutoffs (between IPA and $G_{\max} = 2000$ mHa) should therefore be added to Supplementary Table IV (note also that the main text uses the notation G_{\max} , not G). Just by interpolating these two points, it seems that a smaller cutoff $G_{\max} < 2000$ mHa would obtain an SC similar to the ML models with only modest computational overhead over the IPA. The authors should critically comment to which extend this result relativizes the efficacy of their ML schemes. Notwithstanding this point for the ML of RPA spectra, TL remains an attractive option for the prediction of more advanced spectra (BSE or experiment).

Ad B3) Supplementary Table III: Do the authors have an explanation why the training set losses increase for TL for large training set sizes, while this is not the case for DL? Perhaps some of the TL hyperparameters such as the L2 Regularization are not chosen optimally here. Alternatively, this could indicate overfitting of DL to k-point artifacts, as discussed by the authors in their response to point B5. If the latter is the case, could this overfitting be demonstrated explicitly?

Test set losses should be added to Supplementary Table III for easier direct comparison. For the sake of completeness, SC values for training set and test set should also be added.

Finally, I wonder whether the zero L2 regularization stated for DL (training-set size 300) is a typo in Supplementary Table II.

B8) New minor point, concerning the update in the author response letter: The authors should add some brief Supplementary Information concerning the training of the SC predictor, and state the newly found optimal hyperparameters.

Stefan Riemelmoser, EPFL

(Remarks on code availability)

Reviewer #2

(Remarks to the Author)

Thank you for addressing all my comments and questions in such great detail. The new interactive Plot is a very nice solution to provide more clarity about the dataset.

(Remarks on code availability)

Reviewer #3

(Remarks to the Author)

I thank the authors for addressing the reviewer comments and making appropriate changes.

(Remarks on code availability)

Reply

Letter to the Reviewers

Dear reviewers:

First of all, we would like to thank you for your time and effort in reviewing and handling our manuscript. The three very detailed and very knowledgeable reports addressed issues on a wide range of topics from technical details of figure captions to suggestions of future research directions. They surely helped to improve the manuscript.

We considered all comments (reproduced in green) and answer to all concerns point by point. Responses to each comment are provided below, and changes to the manuscript are emphasized in bold. Changes are marked in the manuscript by color. We also took the opportunity to make minor fixes, such as correcting typographical errors.

In addition, we have noticed that the hyperparameters for the model which predicts $SC[RPA;IPA]$ were chosen very poorly—specifically, they were accidentally set to the transfer learning defaults instead of the direct learning defaults. Correcting this has noticeably improved the (already good) performance. We have included a new figure, which has changed the numbering of all figures. For consistency and to avoid confusion, we refer to figures by their *old* numbering in this reply.

Response to Reviewer #1

The authors extend their GAT-ML scheme for optical spectra (Ref. 12 in the manuscript), by moving from GGA-IPA to GGA-RPA. To improve the efficiency of the ML training, the authors explore a transfer learning approach. Convincing evidence is presented that transfer learning from IPA to RPA outperforms direct learning of RPA spectra. Furthermore, the authors demonstrate that their ML scheme can predict optical spectra of larger systems not included in the training set. Finally, the authors address the "learnability" of RPA spectra, and try to predict the difference IPA/RPA through an additional ML scheme. Overall, the study is well sourced and conducted on a high technical level. The introduction is well written and clearly outlines the need for ML of RPA spectra. The results are clearly presented, and the conclusions drawn are logically sound. To further improve the quality of the manuscript, I propose the following points:

- A1) The authors present a large database containing thousands of RPA spectra for bulk solids. To the best of my knowledge, no such database exists so far, and thus these calculations alone constitute a significant achievement. Given that the inclusion of local-field effects in the RPA is

the main advancement of this work, more effort should be spent on postprocessing and analysing the RPA data. For instance, the importance of local-field effects [as measured by $SC(IPA,RPA)$] can be analysed as a function of crystal structure and correlated to properties of the constituent elements. Here, the authors could make good use of their descriptor presented in Fig. 5, and attempt to reverse-engineer the atomistic origins of strong local-field effects. To support their analysis, the authors could also present a plot similar to their Fig. 1, demonstrating weak, medium, and strong local-field effects. Further, the importance of local-field effects can be correlated to scalar dielectric properties following for example Ibrahim and Ataca (see Fig. 5 in Ref. 13 of the present manuscript). Finally, one can go beyond analysing $SC(IPA,RPA)$ by looking at differences in the position and heights of the main peaks, or by resolving $SC(IPA,RPA)$ to different spectral ranges (IR, visible, UV).

We would like to thank the reviewer for the endorsement of our database. We agree that further investigation of the RPA is warranted, but our main focus is of course on getting to higher levels of theory, so that we can train models to experimental accuracy in the next step.

Though we think a full answer to the question "Which materials need the RPA, and for which is the IPA enough?" is out of scope for this manuscript (and likely an entire project on its own), we have found it especially intriguing and investigated it a bit for this manuscript. While it is easy to find materials with extreme dissimilarity between the IPA and RPA, it is difficult to find commonalities among those with large similarity. In addition, we have visualized the latent space of the SC prediction network (as we have done for spectra prediction networks in previous work) and found that the most 'dissimilar' materials share common chemical motifs (nitrates, azides, materials with an unusually high amount of halogenides). On the other hand, the most 'similar' ones are quite spread out.

To allow for further investigation also by other groups, e.g., along the ideas put forward by the reviewer, we have decided to open-source not just the database, but also the calculation folder of our workflow, which also includes charge densities and intermediate convergence steps.

Corresponding changes and additions in the manuscript, located by line numbers:

Lines 232-256: Which materials are [...] (see Data Availability section).

Lines 267-268: We make the RPA database freely available and estimate that many more physical insights can be gained from a more detailed analysis of it.

Lines 311-316: The UMAP in Fig. 6 [...] if we do not do this.

Lines 319-321: In addition, we also supply [...] intermediate values of G_{max} .

Addition of Figure 7 (new numbering)

- A2) Figure 3: The authors demonstrate that ML accuracy surpasses the minimum baseline of the IPA itself. Given that the IPA can be interpreted as maximally truncating the basis set (lines 70-72), I suggest a more stringent control: RPA spectra with reduced basis set size. The accuracy of the RPA spectra can be plotted as a function of basis set size, and this convergence plot can be compared to the learning curves given in Fig. 2. The method section states that the RPA spectra are obtained iteratively starting with smaller basis sets (lines 210-211). Thus, the data points for the convergence plot should be readily available. In this way, the compute cost of the ML approach (creating the ML database and training the models) can be contrasted with the computational overhead of simply calculating the RPA spectra. Perhaps a concrete case can be made for the larger structures in the test set (7-8 sites in the unit cell, see Fig. 2).

It is a good idea to use RPA with a smaller basis size as a more stringent baseline/control, as that might be a setup used in future high-throughput studies. As materials converge at different points, we have decided to use a constant value of $G = 2000$ mRy to compare against. We report this and time comparisons (see below) in the SI, and as can be seen, RPA spectra with $G = 2000$ mRy are already converged for most materials, though one of course needs to carry out calculations with larger values of G to verify. We have not included it as a baseline (though arguably it is more closer to an upper limit than a baseline) in Fig. 2, as we thought the figure would be too crowded after we had already included the IPA baseline.

Regarding time comparisons:

We have included the total CPU hours used for the generation of the database as well as some numbers for the time per forward pass of the ML model.

A more detailed time comparison is more complex, as different materials need different numbers of convergence steps and, depending on the symmetry of their structure, need 1-3 separate calculations for the inequivalent diagonal elements of the dielectric tensor. In addition, a real comparison needs a set algorithm to compare against. For example, we assume many high-throughput workflows would simply set a constant, small value for G which is good enough for most materials, e.g., $G = 2000$ mRy. One might also already have the necessary files, i.e., charge densities and YAMBO databases, from other calculations as part of a larger high-throughput database. In that case, one would only have to perform the actual RPA calculations. We have therefore evaluated

the time solely the RPA calculations with $G = 2000$ mRy in all symmetry-inequivalent directions take, and find a mean (median) time per material of 3.7 (1.52) CPU hours, around a sixth of the time taken by our full convergence workflow. By comparison, the equivalent IPA (i.e., $G = 0$) calculations take a mean (median) of 6.8 (4.3) CPU minutes. The bottleneck for the latter is usually the preceding DFT calculation using a very fine k-point grid. In summary, a significant amount of computational time is required upfront to generate the training databases. However, processing new materials through a machine learning model is essentially 'free' afterwards. We have also included some remarks on this in the discussion.

Addition of Supplementary Table IV

Lines 289-293: The vast majority [...] median being 7.8 hours.

Lines 263-265: As can be seen [...] is essentially 'free'.

- **B1) Introduction:** machine learning should be discussed in comparison to other strategies aiming to reduce the computational cost of TDDFT/BSE. Such strategies encompass real-time propagation (see e.g. doi: 10.1103/PhysRevB.67.085307), the use of the Lanczos algorithm (see e.g. doi: 10.1103/PhysRevLett.96.113001), and basis set reduction (see point A2). The scaling of these approaches can also be contrasted with ML (see lines 133-136).

This is a good point that should be mentioned for a wider audience, and we have included a brief discussion on more efficient methods in the introduction.

Lines 39-43: We note that better scaling [...] active field of research.

- **B2) Figure 1:** The names of the 9 materials should be given in the plots. Is there any explanation why some spectra are easier/harder to predict for ML (see also point A1)?

Right now, we have no reliable proof, nor even an educated guess, as to why some materials are easier or harder to predict. We found in our earlier work that anecdotally the materials with the worst prediction errors are 'unlike' all materials in training set, but the model treats them reliably like superficially similar materials (e.g. solid fluorine [the only solid involving neutral fluorine atoms] like fluorides [containing negative fluorine anions] or layered hexagonal boron nitride to three-dimensional modifications). The present work shows that RPA and IPA can be predicted similarly well for a given material. Thus, our working hypothesis is that there is an underlying concept of 'similarity between materials', as we discuss in the manuscript, and we are now actively conducting research in this direction.

We have updated the figure to include the chemical formula of the compounds.

Modified Figure 2 (new numbering)

- B3) p. 4: The authors discuss SC and MSE as error metrics (Eqs. 5 and 6). In their previous work (Ref. 12 in the manuscript, SI), the authors indicate that a L_1 loss is used for training. It should be clarified, whether this is still the case (Supplementary Table 1 suggest an L_2 loss is used). If L_1 is still used, L_1 losses should be defined on p. 4 and L_1 losses should be given as addition to errors shown in Fig. 2 (can be added as supplementary). Further, training set errors should be given as supplementary information, too.

Thank you for pointing this out: We still use an L_1 loss, as we now note in the Methods section (we generally prefer the use of the word 'MAE' instead of the word ' L_1 loss', except when specifically referring to the loss function). Supplementary Table 2 shows the optimal coefficient for L_2 regularization, also known as weight decay. We added a definition for the L_1 loss/MAE and have now included training set errors (both MAE and MSE) in the Supplementary Information.

Lines 146-147 and equation 8: During training, we use the L_1 loss [...].

Lines 304-305: All models are trained [...] Supplementary Table III.

Addition of Supplementary Table III

- B4) Figure 2: The authors should indicate IPA as baselines in the learning curves. If this overwhelms the plots, mean IPA errors can also be stated in the text (line 155).

This is a good suggestion, we have added this baseline to Fig. 2.

Modified Figure 3 (new numbering)

- B5) Figure 2: For very large training sets, direct learning seems to outperform transfer learning slightly. This is also discussed by the authors in lines 176-179. However, the authors discuss that SC of 0.9 represents the numerical accuracy of their spectra (line 211). Thus, it is conceivable that DL overfits and learns numerical noise in the training data. This could be tested explicitly (i) by calculating higher accuracy spectra, or (ii) by training DL and TL on lower accuracy spectra.

The reviewer raises an interesting good point. While our convergence threshold (in regards to G-vector convergence) is an SC of 0.9, the median SC between the final and the penultimate calculated spectra is around 0.97 (see also our answer to question A2). We do expect that some overfitting to numerical artifacts is occurring, but we believe that the more significant numerical

artifact are still slight k-point ripples, even though we use extremely dense k-point grids. We have noted the mean convergence SC in the Methods section.

Lines 289-290: The vast majority [...] (over all directions).

- B6) Lines 162-175: The authors discuss the puzzling finding that RPA is as easy to learn as IPA. This could be due to the fact that the database does contain neither transition metals nor low-dimensional materials, where local-field effects would be more pronounced. In future work, it will be interesting to see, whether ML of BSE spectra is equally as easy. In truly non-local cases such as long-range charge-transfer excitons, the current ML scheme seems bound to fail due to the real-space cutoff in the model.

We agree with the reviewer, and we are looking forward to how well our approach will work for a larger set of elements and BSE, topics which we are both working on currently. Regarding truly long-range charge-transfer excitons, we also agree with the reviewer. For the systems we are currently most interested in, relatively small-cell crystalline materials (and not, e.g., organic crystals), our current approach should hopefully perform well, as we have a receptive field of 10 Å already with two message passing operations. In addition, we note that our softmax-attention based pooling is also a fully non-local operation [1].

- B7) Abstract: There is a grammatical error in the first sentence of the abstract, "approximation" is erroneously written twice. Title: "Jacob's Ladder" should be capitalized in the title. Line 185: the authors train a ML model, not a material.

We thank the reviewer for noticing this and have made the corrections.

- C1) p. 1: The authors discuss RPA as the second step on the Jacob's ladder of optoelectronic properties. To convey the importance of their work to a broader audience, the authors could compare IPA, RPA, BSE spectra, and experimental spectra for a select number of materials. Then, the similarity coefficient can be used to contrast $SC(IPA,RPA)$, $SC(RPA,BSE)$, and $SC(BSE,expt)$ (i.e. how big are the steps on the ladder). I recognize that BSE calculations are expensive, but perhaps the authors can take BSE spectra from the literature. Moreover, the presentation of the work might benefit from a small comic illustrating this Jacob's Ladder of optoelectronic properties (similar to the original one, see Ref. 1 in the manuscript).

The reviewer makes a good point, especially for targeting a wider audience—after all, the basic assumption of many people not directly involved in *ab initio* solid-state physics is "DFT cannot get

spectra right”, which is true if one sticks to PBE-IPA, but generally GW-BSE spectra are pretty close to the experiment. One challenge in the proposed recommendation is that SC(IPA,RPA), SC(RPA,BSE) and SC(BSE,expt) vary widely from material to material, as illustrated by the distribution of SC(IPA,RPA) shown in this work—as the reviewer rightfully points out in his other remarks, discovering which method works how well for each material is a research question that is just now becoming possible to answer. We have added a reference to a nice overview for prototypical semiconductors/insulators, but note that generalizing from the simple materials usually analyzed to more exotic compounds is tricky.

The idea to add a cartoon is a good one, and we have included one as well.

We have also included TDDFT as an additional rung between RPA and BSE—after all, the RPA can be considered as “TDDFT with just the Hartree kernel”, and to our knowledge the best available xc-kernels are based on reconstructing the BSE [2, 3].

Addition of Figure 1 (new numbering)

Lines 27-29: A good comparison [...] are analyzed there.

Lines 33-37: To the best of our knowledge, [...] as summarized in Ref. 14.

- C2) Fig. 4 contains somewhat redundant information and could be omitted (cf. Figs. 2 and 3 and Supplementary Fig. 1).

We agree that the information shown in Fig. 4 could be gleaned from Figs. 2 and 3 and the Supplementary Information, but we have decided to keep it in the main manuscript as the manuscript is not exceedingly long anyway and it is, in our opinion, a nice demonstration that direct learning and transfer learning arrive at very similarly good predictions for each material.

- C3) lines 198-202: The authors discuss that their transfer learning scheme could be applied to enable the prediction of GW-BSE spectra. Perhaps it is beneficial to introduce an intermediate step that promotes GGA-IPA spectra to GW-IPA spectra. In this context, for instance the recent work of Knøsgaard and Tyghesen (doi: 10.1038/s41467-022-28122-0) could be useful. As an alternative to GW-IPA, meta-GGA could also be viable (e.g. Ibrahim and Ataca, Ref. 13 in the manuscript).

We agree and have added some sentences regarding this to the conclusion. We are currently finalizing a QSGW-BSE workflow and will most likely generate QSGW-IPA/RPA spectra as intermediate steps. It will be interesting to find out whether stacked transfer learning, e.g. PBE-

IPA to PBE-RPA to GW-RPA to GW-BSE, works better or worse than directly going from PBE-IPA to GW-BSE.

Lines 274-277: It remains to be seen [...] transfer learning step to GW-BSE.

- C4) Lines 183-190: The authors discuss that their similarity descriptor (IPA/RPA) could be used for high-throughput studies. Once future work has extended the current scheme to BSE, an interesting prospect is to adapt the similarity predictor for ML of exciton binding energies.

Yes, we fully agree, and in fact we are already working on BSE. We have previously developed a physics-based estimator for the exciton binding energy [M. Grunert et al., Predicting exciton binding energies from ground-state properties, Phys. Rev. B 110, 075204 (2024)] and are looking forward to see how it will compare to machine-learned predictions.

- C5) Outlook: The prospect of combining ML interatomic potentials with ML optical spectra seems exciting. This could enable fast prediction of optical spectra of large and dynamic systems (e.g. liquids).

This is a good point and a natural suggestion—we have added this is a proposed direction for future studies to the conclusion/outlook.

Lines 258-261: The overall advantage [...] via molecular dynamics.

In conclusion, the study advances the emerging field of ML optical spectra through two main achievements: (i) The construction of a large database of RPA spectra, and (ii) the ML of RPA spectra with a transfer learning approach. In my opinion, not enough emphasis is yet given to achievement (i). The database presents a unique opportunity to explore the physics of local-field effects (see point A1). The authors have also indicated that the database will be publicly released. This will be highly useful for other researchers, and it will contribute to the long-lasting impact of this work. Transfer learning per se [achievement (ii)] is not as novel, but the authors give due credit to prior work. Still, the results are very encouraging. In point A2, I suggest a more challenging control for the ML approach. Such analysis could help to convince potential users to adopt the present scheme. Overall, the present study presents an important step towards ML of GW-BSE spectra, and I support publication in nature communications once the comments have been addressed.

Response to Reviewer #2

The paper reports the results of the fine tuning of a machine learning model predicting the optical spectra

of a large dataset of crystals. The fine tuning involves going from a model trained on the independent particle approximation (IPA) to the results of a random particle approximation (RPA) approach, which in general provides better accuracy when comparing the computed spectra with experiment. Multiple experiments including the comparison of direct learning (DL) vs transfer learning (TL) are conducted to show and compare the performance of their model.

In general, the manuscript is very well written and explains the necessary theory and terms used nicely. The conclusions are clear and well evidenced. The experiments are well designed. The results are noteworthy as they demonstrate how little data of higher level theory for optical spectra is necessary to obtain a model that produces a similar high level of theory, even though similarly little data was used in other transfer learning tasks e.g. for when training machine learning interatomic potentials.

However, please address the following points for clarity:

- For me, the target of the machine learning model is not quite clear. It seems to be the trace of the imaginary part of the dielectric function (as plotted in Fig. 1) or is it the whole imaginary part of the dielectric function as stated in the machine learning methods section? I could not find where this is explained further. Is the real part of the dielectric function also computed and if so, how is the coupling between the imaginary and the real part handled?

The output of the models presented here is the trace of the tensor of the imaginary part of the dielectric function from 0 eV to 20 eV in 10 meV steps, i.e., 2001 separate values. We have perhaps overly relied on references to our earlier work, which describes the model in more detail. We have now included a more detailed description of the model in this manuscript as well.

Currently, we use separate models to predict the real and the imaginary part. We have experimented with models which predict both at the same time, but found no significant improvement. We also experimented with physically-inspired loss functions by adding an auxiliary training loss of the difference between the real part of the dielectric function and the Kramers-Kronig transformed imaginary part (and vice versa), but found no significant improvement either. In general we found that—with no intervention on our part—the real and imaginary part fulfill the Kramers-Kronig relations very well already [to be published].

In this work, as we were primarily interested in the efficacy of transfer learning, we focused solely on the imaginary part.

Lines 106-114: The model is sketched in [...] 0 eV to 20 eV.

Lines 116-117: In this work, [...] dielectric tensor $\varepsilon(\omega)$.

Addition of Supplementary Figure 2

- Could you provide in the SI the elemental compositions of your dataset? The diversity of the dataset is not clear to me. How many elements are involved? Are these only crystalline or also amorphous structures? High fidelity is claimed but no exact description is given.

We have included more descriptions of the dataset in the main manuscript, and have added a visualization of the elemental diversity of the dataset as Supplementary Figure 3. The dataset consists of a subset of materials of the Alexandria database [4], which itself consists only of ordered crystalline materials.

High fidelity data in the specific way used in the manuscript is used as in the literature on transfer learning or multi-fidelity machine learning, i.e., to signify better quality but less readily available data (in this case, data at a higher theory level). We agree that this is confusing to readers not familiar with that specific field and have accordingly added a sentence making this clear.

Line 54: In machine-learning literature, such more accurate but more costly data is usually referred to as high-fidelity data.

Lines 101-105: The data spans [...] Data Availability section.

Addition of Supplementary Figure 3

- Figure 1: what does the small dataset in the caption refer to? Is this the cell sizes or the dataset size? Also, a legend to easier compare the data would be nice. Also, what role does the scaling in y-direction of the spectra play here? When scaled the same, the IPA and RPA spectra for Q10 would be basically the same. Does it make sense to compare the absolute values of the spectra and not scale them to have the same integral first? It would also be nice if the chemical structure of the calculated materials would be given, so a bit more information on the dataset would be obtained. Also, what are the proportions of the different quantiles in the dataset (just to estimate their influence on the training/testing)?

The bold description of Figure 1 was referring to a previous version of the figure, and we thank the reviewer for bringing this to our attention. Figure 1 shows the performance of both strategies after training on different dataset sizes, which we have made clearer in the fixed description. We have also added a legend, and added the chemical formulas of the compounds shown. We have not added images of the crystal structures, as we felt it overwhelmed the plot, but we note that the interactive version of the new Figure 7 shows the crystal structure for each material.

It would be possible to normalize the dielectric function, however one would also have to learn the normalization factor as a separate value. E.g, the absolute value and the renormalization that occurs for $Q_{10\%}$ is also physically meaningful. This would amount to a different representation of the dielectric function, and is the way Ibrahim and Ataca [Ref. 19 of revised manuscript] have done it. For the transfer learning in this work, we need to keep the underlying base model, which was optimized in detail already [1]. Regarding different representations, we had initially experimented with a variety of those, but did not find that any perform significantly better than any other. This agrees with private communications we had with other groups working in this field. Perhaps it would be a good idea to investigate this again in more detail and publish it, even in case no difference in the performance of different representations could be found.

Modified Figure 2 (new numbering)

- Please explain what the trace of the imaginary part of the dielectric function under IPA relates to, is this a simple adsorption spectrum?

The imaginary part of the dielectric function is related to, but not equal to an absorption spectrum. We have added a small discussion on why we focus on the dielectric function in particular, and how it relates to absorption spectra, which are probably the other main optical property usually measured.

Lines 116-122 and equation (5): In this work, [...] e.g., using ellipsometry.

- Page 4: 'consisting of only the materials with at most 2, 3, 4, 5 and 6 sites per unit cell'. What are sites? Unique crystal sites or elements or elements on a specific crystal site?

We agree and regret that we have been unclear here. Whenever we referred to "site", we meant "atom in a primitive unit cell", i.e., elemental silicon has 2 "sites". We have changed the wording to make it clearer that we are referring to number of atoms in the primitive cell.

Changed 'sites' to 'atoms per primitive unit cell' at various places in the manuscript.

- Is the transfer learning on small cells evaluated on the full test set with also more sites in the cell? How large were the test sets?

We always evaluate on the full test set, except for Fig. 2 (bottom), where we split the full test set into smaller sets based on the number of atoms per primitive cell, to show that transfer learning on materials with only few atoms per primitive cell generalizes to materials with more atoms per primitive cell. The number of materials in each of these subsets is given by the background

histogram in Fig. 2 bottom. We have made the caption of Fig. 2 clearer and we have added the number of materials in the full training/validation/test sets.

Caption of Figure 3 (new numbering): Boxplot of the distribution [...] in the test set.

- **Figure 3: why is ad plotted?** It does not make sense to compare the direct model with the similarity of the IPA and the RPA spectra.

The reviewer is correct, the plot alone does not make much sense. It is shown only because it naturally results from the intersection of the other plots we show. It is certainly a matter of taste, but we like the composition of Fig. 3 and Supplementary Figure 1 as a visually appealing way to present several one- and two-dimensional histograms.

- **Learning the similarity:** The motivation is good, however why are no values given for which kind of material classes are adequately computed with IPA and which require RPA? Are there trends visible? This information would add to the scientific message of the paper.

This is a good point, and we have added a section analyzing this. We also refer to our answer to question A1 by reviewer #1.

Lines 231-256: Which materials are [...] (see Data Availability section).

Lines 267-268: We make the RPA database freely available and estimate that many more physical insights can be gained from a more detailed analysis of it.

Lines 311-316: The UMAP in Fig. 7 [...] if we do not do this.

Addition of Figure 7 (new numbering)

Response to Reviewer #3

The authors present a framework for predicting the optical spectra of semiconductors using graph neural networks and transfer learning from the lower-fidelity IPA calculations to the higher-fidelity RPA calculations. It is shown that while there are obvious differences between IPA and RPA estimates, transfer learning does help improve the ML prediction of the RPA-level optical spectrum. Analysis also shows that models can be trained mostly on smaller system sizes and extended to larger systems. The DFT dataset made available in this work will be very valuable.

I believe the work is strong, but a number of things must be clarified and changed before the manuscript is accepted for publication.

- 1. There is reference to the earlier publication that reports the DFT calculations and dataset, but I believe it would be helpful if the authors added some sentences about the chemical composition of the dataset. What kind of compounds are being used here? Does this contain oxides, halides, chalcogenides, etc., or is it restricted to particular classes of compounds? What role does chemistry play in this work?
- 2. It is also not clear to me how the GNN models are being trained- the crystal structure should be the input, and is the output the entire dielectric function spectrum? Is it broken into bins for prediction?
- 3. What specific GNN algorithms were used in this work, and which method works best?

We thank the Reviewer for these comments. We agree that we were very brief, possibly too brief, in the description of both the model and the used dataset. We have added a section describing both, and have also added Supplementary Figures showing the elemental diversity of the used data set and the architecture of the used model.

Regarding point 3: The basic architecture of the model as described in the article was kept constant from the original publication to allow for a fair comparison. In the original publication, a detailed hyperparameter study was carried out, and the base architecture was optimized via trial-and-error. We are currently in the process of carrying out an in-depth optimization and further development of the architecture on another dataset, but will not go into this any further in the present manuscript in order to guarantee maximum comparability.

Lines 101-114: The data spans [...] 0 eV to 20 eV.

Addition of Supplementary Figures 2 and 3

- 4. Figure 1 is slightly confusing without a legend—I think something should be added to the right side of the figure to show what each color and type of line represent.

We have added a legend to the right of the figure.

Modified Figure 2 (new numbering)

- 5. Related question on Figure 1—why is the solid green line alone very squiggly compared to the other lines?

The solid green line is so squiggly because it is the output of the model trained directly on only 300 datapoints—too few for the model to learn that dielectric functions should be smooth. All

other models have enough training data (4610 RPA materials for the dashed green line, 7743 IPA materials and 300 RPA materials for the solid orange line, 7743 IPA materials and 4610 RPA materials for the dashed orange line). In the unrevised manuscript, we have only described the solid green line as being "unphysical". We have added some more descriptions of why this is the case and what exactly we meant with "unphysical".

Caption of Figure 2 (new numbering): on 300 materials [...] were used for training.

- 6. It would also be useful to add actual parity plots showing ML prediction vs DFT in the SI.

It is not straightforward to create parity plots for spectral data, as each data point is multidimensional and it is unclear how to reduce it to a scalar for the creation of parity plots. We have decided to evaluate (an approximation of) the quantum weight, i.e., the integral of the spectrum over the calculated frequency range, and use that for a parity plot, which is now included in the SI.

Addition of Supplementary Figure 4

- 7. Can the authors present a time (or computational expense) comparison between full DFT and the TL-based prediction?

On the DFT side, we have now included both the total computational expense for generating the database, and the computational time necessary solely for the optical calculation. For the ML prediction, we now also report the time required for an entire prediction, taking into account reading in of a CIF file as a PYMATGEN structure object (which is the bottleneck, if you want to call the slowest step that), its conversion to a graph, and the execution of the forward pass (see also our reply to question A2 of Reviewer #1).

Lines 291-293: In total, the *ab initio* calculations [...] median being 7.8 hours.

Lines 305-310: Training a machine learning model [...] running the forward pass (3.51 ± 0.1 ms).

-
- [1] M. Grunert, M. Großmann, and E. Runge, Deep learning of spectra: Predicting the dielectric function of semiconductors, *Phys. Rev. Materials* **8**, L122201 (2024).
- [2] G. Onida, L. Reining, and A. Rubio, Electronic excitations: density-functional versus many-body Green's-function approaches, *Rev. Mod. Phys.* **74**, 601–659 (2002).

- [3] Y.-M. Byun, J. Sun, and C. A. Ullrich, Time-dependent density-functional theory for periodic solids: assessment of excitonic exchange–correlation kernels, *Electronic Structure* **2**, 023002 (2020).
- [4] J. Schmidt, N. Hoffmann, H. Wang, P. Borlido, P. J. M. A. Carriço, T. F. T. Cerqueira, S. Botti, and M. A. L. Marques, Machine-learning-assisted determination of the global zero-temperature phase diagram of materials, *Adv. Mater.* **35**, 2210788 (2023).

Reply

Letter to the Reviewers

Dear reviewers:

We would like to thank you for your time and effort and especially your speed in reviewing our manuscript. We would also like to thank Reviewer #1 for his attention to detail and his consistent efforts to improving our manuscript, which we believe have cleared a few remaining ambiguities.

We considered all comments (reproduced in green) and answer to all concerns point by point. Responses to each comment are provided below, and changes to the manuscript are emphasized in bold. Changes are marked in the manuscript by color.

Response to Reviewer #1

The authors have addressed most of the points raised in an extensive and satisfactory manner. In particular, the additional analysis on the physics of the RPA is interesting (new Figure 7, see author response to point A1). I also welcome the decision of the authors to add supporting data to their publicly available database. A few minor issues still warrant further response, however:

- Ad A2) Supplementary Table IV: I appreciate the clarifications concerning computational cost brought here and in the main text. The cutoff chosen by the authors ($G_{\max} = 2000$ mHa) is indeed an upper limit, given that no ML models actually reach $SC = 0.97$. If the data is available, 2-3 more points at intermediate cutoffs (between IPA and $G_{\max} = 2000$ mHa) should therefore be added to Supplementary Table IV (note also that the main text uses the notation G_{\max} , not G). Just by interpolating these two points, it seems that a smaller cutoff $G_{\max} < 2000$ mHa would obtain an SC similar to the ML models with only modest computational overhead over the IPA. The authors should critically comment to which extend this result relativizes the efficacy of their ML schemes. Notwithstanding this point for the ML of RPA spectra, TL remains an attractive option for the prediction of more advanced spectra (BSE or experiment).

The reviewer is correct that a smaller cutoff would in most cases still suffice. Unfortunately, we have no way to verify or quantify this, as we have used a (in hindsight) relatively large step size of 2000 mRy, and most materials are converged for $G_{\max} = 4000$ mRy. The step size was originally taken from our previous work on converging G_0W_0 calculations, for which one step is calculating $\varepsilon_{\text{RPA}}(\mathbf{q}, \omega)$ (though under the used plasmon-pole approximation, the dielectric function is only calculated for two values of ω). In that work, we generally needed significantly higher values of

G_{\max} to reach convergence for the G_0W_0 band gap [1] than in this work for the convergence of the optical spectra. We do not know why this is the case. Potentially, we might have used more stringent convergence criteria (it is not clear how to translate convergence in the band gap to convergence in the overall spectra), or either the $\mathbf{q} \neq 0$ -components of the dielectric function or the self energy itself might converge slower with respect to the number of \mathbf{G} vectors. This is an aspect that we think could be interesting to investigate in future work, and we have added a sentence pointing this out. In addition, we have also explicitly noted, when discussing our choice of step size, that smaller step sizes are likely a better choice for future high-throughput work. We would also like to thank the reviewer for pointing out the inconsistent notation.

Updated notation in Supplementary Tab. IV

Lines 289-292: It is reasonable to expect [...] clear to us.

- Ad B3) Supplementary Table III: Do the authors have an explanation why the training set losses increase for TL for large training set sizes, while this is not the case for DL? Perhaps some of the TL hyperparameters such as the L_2 Regularization are not chosen optimally here. Alternatively, this could indicate overfitting of DL to k-point artifacts, as discussed by the authors in their response to point B5. If the latter is the case, could this overfitting be demonstrated explicitly? Test set losses should be added to Supplementary Table III for easier direct comparison. For the sake of completeness, SC values for training set and test set should also be added. Finally, I wonder whether the zero L_2 regularization stated for DL (training-set size 300) is a typo in Supplementary Table II.

We have optimized (separately for all training set sizes and learning strategies) the L_2 -parameter, the learning rate and the batch size, and for the DL case also the architecture parameters. We have found that in general, the optimal DL models have more parameters when going towards larger datasets, which is to be expected. For the TL case, we have stuck with the original OPTIMATE model, and it is possible that it is running out of model capacity for the larger training sets and is less likely to overfit the training set than the DL. Due to computational restrictions at the time, the original OPTIMATE model is smaller than the largest of the DL models in this work. On the other hand, training set losses are a bit tricky to interpret anyway, as they are competing against minimizing the L_2 regularization. As we have no good answer to this, we have decided not to discuss it in the manuscript.

We have added test set losses and SCs to Supplementary Tab. III.

The L_2 regularization of 0 is not a typo. However, we generally find that the impact of L_2 regularization is usually not large, and similar results are obtained over a relatively wide range. For demonstration, the table below shows the best five sets of optimizer hyperparameters for DL on 300 datapoints. The training is apparently not very sensitive to the exact choice of hyperparameters.

TABLE I. Performance and hyperparameters for DL on a training set of 300 materials. η signifies the learning rate and L_2 signifies the L_2 -regularization parameter. Architecture hyperparameters are chosen as in Supplementary Tab. I. Error measures are obtained as averages from models trained via 5-fold CV on the 300 materials training set and evaluated on the validation set. The table is sorted according to median SC and shows the five optimal sets of hyperparameters.

Mean MSE	Median MSE	Mean SC	Median SC	η	L_2	Batch size
1.033284	0.254190	0.284249	0.754999	0.00001	0.000000	20
0.947824	0.273215	0.282717	0.754653	0.00010	0.000000	20
0.960850	0.277051	0.280789	0.754382	0.00010	0.000100	20
0.964531	0.278434	0.287157	0.753806	0.00010	0.000100	64
0.951947	0.271580	0.287918	0.753770	0.00010	0.000001	64

Regarding k-point overfitting, we think it could be demonstrated, but doing so would likely be a project in itself. For the original IPA dataset, spectra at different k-point grid densities are available. One could—in principle—train a model on coarser k-point grid densities and see if one ends up overfitting towards "k-point ripples". However, we think this would go beyond the scope of the current work.

Updated Supplementary Tab. III

Line 308: and test set

- B8) New minor point, concerning the update in the author response letter: The authors should add some brief Supplementary Information concerning the training of the SC predictor, and state the newly found optimal hyperparameters.

Thank you for noticing this, we had forgotten to mention the hyperparameters in the text. The main change to the model in the original manuscript was that the learning rate was set unreasonably low. We did not separately optimize the hyperparameters for the SC prediction model, as

we are happy with the performance as is. We have now added a new, very short Supplementary Note 5 detailing the parameters.

Addition of Supplementary Note 5

Lines 225-226: (see Supplementary Note 5 for training details)

- [1] M. Großmann, M. Grunert, and E. Runge, A robust, simple, and efficient convergence workflow for GW calculations, *npj Comput. Mater.* **10**, 135 (2024).